# Sensory and Physicochemical Quality, Residual Fungicide Levels and Microbial Load in ‘Florida Radiance’ Strawberries from Different Disease Control Treatments Exposed to Simulated Supply Chain Conditions

**DOI:** 10.3390/foods10071442

**Published:** 2021-06-22

**Authors:** Katrina Kelly, Yavuz Yagiz, Zheng Li, Gail Mahnken, Wlodzimierz Borejsza-Wysocki, Maurice Marshall, Charles A. Sims, Natalia Peres, Maria Cecilia do Nascimento Nunes

**Affiliations:** 1Food Quality Laboratory, Department of Cell Biology, Microbiology and Molecular Biology, University of South Florida, 4202 E. Fowler Avenue, Tampa, FL 33620, USA; ktakelly@gmail.com; 2Department of Food Science and Human Nutrition, University of Florida, 520 Newell Drive, Gainesville, FL 32611, USA; yavuzy@ufl.edu (Y.Y.); yzhang@usf.edu (Z.L.); gmahnken@ufl.edu (G.M.); wb17@cornell.edu (W.B.-W.); martym@ufl.edu (M.M.); csims@ufl.edu (C.A.S.); 3Gulf Coast Research and Education Center, University of Florida, 14625 Co. Rd. 672, Wimauma, FL 33598, USA; nperes@ufl.edu

**Keywords:** *Fragaria × ananassa*, supply chain, sensory quality, bioactive compounds, sugars, fungicides, microbial load

## Abstract

Strawberries are greatly appreciated for their flavor and health-promoting properties. However, current agricultural and postharvest handling practices may result in decreased fruit quality. The objective of this work was to determine the effect of conventional or reduced fungicide applications on the quality of ‘Florida Radiance’ strawberries exposed to supply chain conditions. Strawberries held under steady temperature had better sensory and physicochemical quality than fruit exposed to supply chain conditions, regardless of the disease control treatment. Strawberries from the reduced fungicide treatment were firmer, lost less moisture, had higher sugar and higher or similar bioactive contents than fruit from the conventional treatment. Sensory scores were better for reduced fungicide fruit held under steady temperature conditions than other treatments at the consumer level. Microbial load increased during the supply chain but results strongly suggest that washing the fruit significantly reduces the microbial load and residual fungicide levels (fludioxonil, cyprodinil, pyraclostrobin, and captan) on the fruit. Overall, the use of reduced fungicide applications to control strawberry disease constitutes a promising alternative to conventional practices. It will help reduce costs by reducing labor and the amount of fungicides used while maintaining overall strawberry quality. Moreover, avoiding abusive and fluctuating temperature conditions during the supply chain will extend shelf-life and reduce strawberry waste.

## 1. Introduction

Strawberries are amongst the most popular fruits consumed worldwide and are recognized for their exceptional nutritional qualities. Strawberries have been the focus of many studies for their health benefits due to high levels of bioactive compounds, including phenolic acids, flavonoids, and vitamin C. However, because of their perishable nature and mishandling, large quantities of fruit are often discarded throughout the supply chain [1,2,3,4,5,6]. Strawberries that reach consumers and are ultimately consumed fresh may have lost up to 50% of their potential “bioactive quality” due to poor temperature management [7]. Therefore, even though the initial overall quality of strawberries may be high at harvest, a significant amount of the health-promoting compounds is often lost before consumption due to poor handling conditions from the field to the consumer. Poor temperature management, such as delays before cooling, fluctuating, and high temperatures at any point during the supply chain, are significant problems contributing to the loss of quality and strawberry waste [1,7,8,9]. For example, reduction in vitamin C and total phenolic contents at the consumer level can be up to 57% when strawberries are stored or shipped at temperatures above 1 °C as opposed to being kept at a constant optimum temperature [7,9]. When strawberries are exposed to supply chain conditions, sugar and acid contents may decline significantly, resulting in fruit with a more inferior flavor [6,7]. Moreover, softening and the increased loss of moisture because of exposure to abusive supply chain conditions may contribute to the rejection of strawberries due to objectionable appearance at the retail level [1,7,10,11].

Incidence of decay, particularly by *Botrytis cinerea* and *Rhizopus stolonifer*, and survival of microbial contaminants may also increase in strawberries exposed to abusive supply chain conditions from the field to the consumer, resulting in fruit being wasted while also creating a safety risk [6,8,11,12,13,14]. For example, the incidence of Botrytis and Rhizopus fruit rots may increase up to 85% when strawberries are not promptly cooled after harvest [8]. Thus, to reduce field or postharvest diseases, current control treatments, particularly in warm and humid climates such as Florida, involve frequent fungicide applications that may affect strawberry quality [15,16]. Furthermore, consumers are becoming increasingly concerned about the levels of pesticides in their foods. Hence, there is a need to offer alternative agricultural practices that, together with proper handling conditions from the field to the consumer, will produce strawberries with better overall quality. Some studies showed that strawberries grown under organic and reduced fungicide regimes might have superior quality than conventionally grown fruit [15,17,18,19,20,21,22,23]. However, it is unknown how supply chain conditions may affect their overall quality. Studies that compare quality attributes of conventional against organic strawberries may be used to provide some insight. Several have shown that, in general, strawberries organically grown or from integrated pest management (IPM) may have an equal, or superior, quality compared to conventionally grown fruit but have no detectable or reduced pesticide residues [15,17,18,19,20,21,22,23].

To our knowledge, there are no published studies that report the impact of supply chain conditions on the quality of strawberries grown using conventional agricultural methods and the reduced fungicide application method used by our research team [24,25,26]. Thus, the rationale of conducting this work was that by lessening the levels of fungicides used to control strawberry diseases combined with good supply chain practices from the farm to the consumer, overall postharvest quality could be maintained or enhanced. The overall objective of this study was to determine the impact of supply chain conditions on the sensory quality, physicochemical attributes, fungicide residues, and microbial load on ‘Florida Radiance’ strawberries grown conventionally or under the reduced fungicide application.

## 2. Materials and Methods

### 2.1. Plant Material and Disease Control Treatments

‘Florida Radiance’ strawberries were grown in commercial fields in Plant City, FL, USA, under conventional and reduced fungicide disease management conditions using a forecasting system [25,26]. The cultivar ‘Florida Radiance’ (known as ‘Florida Fortuna’ outside the USA.) was chosen because at the time this study was conducted, it was the leading strawberry cultivar grown in central Florida [27,28]. Table 1 shows the types of fungicides used and application dates for conventional and reduced fungicide strawberries grown under the two different disease control treatments.

### 2.2. Postharvest Treatments

‘Florida Radiance’ strawberries were harvested twice during the 2015 production season, on 3 and 10 March. Eight flats of strawberries from the conventional disease control treatment (i.e., 64 clamshells containing approximately 0.453 kg of fruit each; total fruit weight ≈ 29 kg) plus eight flats of strawberries from the reduced fungicide disease control treatment (i.e., 64 clamshells containing approximately 0.453 kg of fruit each; total fruit weight ≈ 29 kg) were brought to the USF-Food Quality Laboratory in Tampa, FL, USA, within about one hour of harvest. From the total number of clamshells, 32 clamshells per disease control treatment were used for the sensory analysis, and 8 clamshells per disease control treatment were used for residual fungicide analysis. From the remaining 24 clamshells from each disease control treatment, 360 strawberries per disease control treatment were randomly selected for uniformity of color and freedom from defects. Thirty-six of these fruit were used for initial quality evaluations. Three replicate samples of 12 fruit, each for the control and per supply chain step, were carefully distributed to three clamshells (capacity ≈ 0.453 kg) and used for non-destructive analysis (i.e., subjective appearance and weight loss). For destructive analysis (color, texture and chemical analysis), three replicate samples of 12 fruit each, for the control and per supply chain, were carefully distributed to three clamshells (capacity ≈ 0.453 kg). The clamshells containing the fruit for the sensory and residual fungicide analysis and both non-destructive and destructive quality evaluations were then stored for specific times inside temperature and humidity-controlled chambers (Forma Environmental Chamber Model 3940 Series, Thermo Electron Corporation, OH, USA Waltham, MA, USA) set at 1.5 ± 0.3 °C, 2.0 ± 0.3 °C, 3.0 ± 1.0 °C, 4.0 ± 0.2 °C, 5.0 ± 0.2 °C, 7.0 ± 0.1 °C, and 30.0 ± 0.3 °C and between 30.0 and up to 85% RH, depending on the supply chain step (Table 2). The fruit’s quality was evaluated at each step individually, after a total supply chain length of 6 days. The total time (6 days) was chosen based on a typical supply chain for strawberry: harvest → cooling → storage at grower → transport from grower to distribution center (DC) → storage at DC → transport from DC to stores → display at the store → purchase by consumer and storage at home. Simulated supply chain conditions within each step were selected based on time–temperature profiles previously measured during strawberry handling [1,6,29]. Strawberry steady storage conditions (1.5 °C and 85% RH; control) were selected based on data from Nunes [30] and Mitcham [31]. Cooling delays (0.13 days) and field temperatures (30 °C) were selected based on data from the Florida Automated Weather Network [32] and on the average field temperatures measured in Florida between January and March [33]. Pre-cooling times and temperatures (2 °C for 0.04 days), storage at the grower’s facilities (1.5 °C for 0.83 days), shipping to the DC (3 °C for 2 days), storage at the DC (4 °C for 1 day), shipping from the DC to stores (5 °C for 0.17 days) and display in the store (7 °C for 1 day) were selected based on published data [1,6,29]. The time (2 days) used to simulate shipping from the grower to the distribution center (DC) was chosen based on the approximate time–distance from Florida to the Midwestern States or Eastern Canada. Finally, conditions used for consumer handling (4 °C for 1 day) were chosen based on typical household refrigerator temperatures [34]. Relative humidity was maintained around optimum conditions (85%) for strawberries [31] except for supply chain steps where previous RH measurements were generally lower [1].

The temperature and relative humidity (RH) were monitored throughout the study using HOBO^®^ brand U12 data loggers (Onset Computer Corporation, Pocasset, MA, USA), which record within an accuracy of +/− 0.35 °C. The RH was monitored using HOBO^®^ brand U12 data loggers (Onset Computer Corporation, Pocasset, MA, USA), which record within an accuracy of ± 2.5% from 10 to 90% RH.

### 2.3. Sensory Analysis

Sensory quality was evaluated at the consumer level after washing the fruit. Fungicide treatments for each temperature scenario (i.e., control and supply chain) and each disease control treatment were subjected to acceptability testing by a panel of 100 strawberry consumers. The panelists were selected based on their consumption frequency of strawberries and availability for all panels. Two strawberries from each treatment were presented to panelists for evaluation. Each treatment was coded by a three-digit random number, and all possible orders of presentation were given approximately an equal number of times. Panelists rated their level of acceptability for overall liking, appearance, texture, and flavor using the 9-point hedonic scale where 1 = dislike extremely, 5 = neither like nor dislike, and 9 = like extremely. Evaluations were conducted in a sensory testing lab with 10 individual booths and a computer data entry system using Compusense^®^.

### 2.4. Instrumental Color and Texture Analysis

Two-color measurements were taken on opposite sides of the fruit in the equatorial region. A hand-held tristimulus reflectance colorimeter (Model CR-400, Minolta Co., Ltd., Osaka, Japan) was used following the procedure described by Kelly et al. [35,36]. The firmness of each strawberry was measured using a TA.XT Plus Texture Analyzer (Texture Technologies Corp., Hamilton, MA, USA) as described by Whitaker et al. [36]. The force required to compress the fruit by 3 mm was recorded in kgf and converted to Newton (N = kgf × 9.8).

### 2.5. Weight Loss and Dry Weight

Weight loss of three replicate samples of 12 strawberries was calculated from the fruit’s initial weight and after each supply chain simulation (6 days). Concentrations of chemical constituents were expressed in dry weight to show the differences between treatments that might be obscured by differences in water content [35]. Chemical compounds were expressed in g kg^−1^ on a dry weight basis to compensate for water loss during storage.

### 2.6. Acidity and Soluble Solids Content

Three replicate samples of 12 individual fruit per treatment were homogenized in a laboratory blender at high speed for 2 min. The resulting puree was immediately frozen and kept at −30 °C until used. Titratable acidity and soluble solids content (SSC) were determined according to Nunes et al. [32].

### 2.7. Sugar Analysis

Frozen samples were thawed at 4 °C overnight, and 0.002 kg of strawberry fruit puree (from each of three replicate samples of 15 strawberries each) was combined with 0.008 L of ultrapure water (Ω 18−17) and then centrifuged at 1811× *g* for 10 min. The obtained supernatant was filtered through a 0.45 µm nylon filter into 0.002 L labeled vials. Quantification of sucrose, fructose, and glucose was conducted using a Hitachi HPLC with a refractive index detector and a 300 mm × 8 mm Shodex SP0810 column (Shodex, Colorado Springs, CO, USA) with an SP-G guard column (2 mm × 4 mm) as described by Kelly et al. [35].

### 2.8. Ascorbic Acid Analysis

Total ascorbic acid was quantified by mixing 0.002 kg of strawberry fruit puree (from each of three replicate samples of 15 strawberries) with 0.02 L metaphosphoric acid mixture (6% HPO_3_ containing 2 N acetic acid). Samples were then filtered (0.22 μm) before HPLC analysis. The ascorbic acid analysis was conducted using a Hitachi LaChromUltra UHPLC system with a diode array detector and a LaChromUltra C18 2μm column (2 × 50 mm) (Hitachi, Ltd., Tokyo, Japan) as described by Kelly et al. [35]. 

### 2.9. Total Phenolics and Anthocyanin Analysis

Total phenolic compounds were measured using the Folin–Ciocalteu reagent as described by Nunes et al. [37]. Anthocyanins were extracted in 0.5% (*v*/*v*) HCl in methanol and measured using the procedure described by Nunes et al. [37].

### 2.10. Fungicide Analysis

Strawberry samples were extracted for multi-residue determination of fungicides based on the QuEChERS method developed by Lehotay et al. [38] and slightly modified by Lesueur et al. [39]. Captan was analyzed using a GC/MS system (6890N GC coupled with an MSD 5973, Agilent Technologies, Santa Clara, CA, USA) with a ZB-5MSi (30 m × 0.32 mm × 0.25 µm) capillary column (Phenomenex, Torrance, CA, USA) under the following conditions: A constant helium flow of 1.3 mL/min; inlet temperature starting at 100 °C and after one min, ramped at 15.2 °C min^−1^ to 235 °C, held 5 min, and ramped at 15 °C min^−1^ to a final temperature of 300 °C withholding time of 5 min. An injection volume of 1 µL in splitless mode, ion source temperature (230 °C), and MS Quad temperature (150 °C). Captan was quantified at 149 *m*/*z* (quantification ion) and 79 *m*/*z* (confirmation ion) with the selected ion monitoring mode. The method was validated at 0.025 and 0.25 ppm captan fortification levels. An HPLC coupled to a mass spectrometer (TSQ Quantum Ultra LC/MS/MS, Thermo Finnigan, Waltham, MA, USA) was used to quantify fludioxonil, penthiopyrad, cyprodinil, cyflufenamid, azoxystrobin, and pyraclostrobin [38,39]. The method was first validated at 0.025 and 0.25 ppm fortification levels on the strawberry matrix before sample analysis. Spike recoveries should fall in the acceptable range of 70 to 120%. Both fortified and unfortified control samples were analyzed concurrently with each sample set to demonstrate the absence of significant interferences and adequate recoveries.

### 2.11. Microbial Analysis

The microbial load on strawberries from each disease control treatment was assessed at three different points during steady or simulated farm-to-consumer conditions (Table 1). The first sample was collected upon arrival of the strawberries to the laboratory from the field (at harvest). The second sample was taken after 4 days, during which the temperature and RH fluctuated to simulate the conditions from harvest to after storage at the distribution center (storage at DC). A final sample was taken at the end of the simulated supply chain (consumer level; 6 days). This was to assess the effect of the disease control treatments and simulated supply chain conditions on the microbial numbers present on strawberries. At the consumer level, the microbial load was determined before and after washing the fruit. Strawberries were washed under running tap water for approximately 1 min, to simulate consumer washing at home. Out of 12 fruit, three strawberries were randomly selected, weighed using an analytical balance (METTLER TOLEDO AL 104, METTLER TOLEDO GmbH, Greifensee, Switzerland), placed inside a Fisherbrand^TM^ lab blender filtered bag with a strainer element (400 mL, 177 mm × 305 mm, 0.5 mm pore size, Fisher Scientific, Waltham, MA, USA) and stomached for 2 min using a Stomacher 400 Circulator (Seward Laboratory Systems Inc., Port Saint Lucie, FL, USA). The bag was then removed from the stomacher and the filter with smashed strawberries discarded. The liquid part was used to prepare a series of 1^−10^ dilutions using test tubes filled with 9 mL of 0.1% peptone water. A 0.001 L sample was withdrawn from each bag and transferred to test tubes (serial dilutions). A volume of 0.001 L from each serial dilution was then plated onto tryptic soy agar (TSA) triplicated plates to detect the microbial load. The plates were incubated at 37 °C for 24 to 48 h. After incubation, the total microbial numbers expressed as a colony-forming unit (CFU) was counted and presented as log_10_ (CFU g^−1^). Each analysis was conducted in triplicate, from which the mean value was calculated. 

### 2.12. Statistical Analysis

The Statistical Analysis System computer package (SAS Institute, Inc., 2004, Cary, NC, USA) was used to analyze the data. The data were treated by two-way analysis of variance (ANOVA) with the harvest, disease control, and temperature treatments as main effects. Significant differences between disease control treatments were detected using the least significant difference (LSD) at the 5% level of significance. Data from the sensory quality and residual fungicides from the two different harvests were analyzed separately. Acceptability data from the sensory analysis were subjected to a two-way analysis of variance to determine whether significant differences exist between the fungicide treatments. If significant differences were indicated, means were separated by Tukey’s HSD (*p* > 0.05).

## 3. Results

### 3.1. Sensory Quality

In the first harvest, the appearance of strawberries from the reduced fungicide treatments (control and supply chain) received better scores in general than fruit from the conventional disease treatment maintained at a constant temperature (Table 3). However, there was no significant difference (*p* > 0.05) between the appearance of fruit from the reduced fungicide treatment kept at a constant temperature (control) and the supply chain treatment. Strawberries from the reduced fungicide treatment at a constant temperature (control) received higher scores for texture, flavor, and overall liking than the other treatments (*p* < 0.05). Although the appearance, texture, flavor, and overall liking tended to be higher for the control fruit from the reduced fungicide treatments, the differences were more subtle in the second harvest. 

### 3.2. Color and Texture

The color of ‘Florida Radiance’ strawberries changed from a light to a deeper red, regardless of the treatment (Figure 1). Although we expected that strawberries at a constant temperatures (control) would have a lighter red color compared to fruit exposed to simulated supply chain conditions, in the first harvest, after 6 days, there was no significant difference (*p* < 0.05) in the L*, a*, and hue angle values of conventional or reduced fungicide strawberries between the control and the supply chain treatments. In the second harvest, there was no significant difference (*p* < 0.05) in the L*, a*, and hue angle values between reduced control and reduced supply chain strawberries; however, the conventional supply chain fruit had a lighter red color (higher L* and hue angle values) compared to the control. The differences between conventional and reduced fungicide strawberries were also not consistent; however, strawberries from the reduced fungicide treatment were on average darker red (lower L* and hue values) than conventional fruit (Figure 1). 

Strawberries from the second harvest were, on average, firmer than fruit from the first harvest; however, they tended to soften more during storage, regardless of treatment (Figure 2). At the end of storage at a constant temperature or supply chain simulation, there was no significant difference (*p* > 0.05) in the texture of strawberries from the first harvest, regardless of the treatment (Figure 2A). However, in the second harvest, strawberries from the reduced pesticide treatment kept for 6 days at a steady temperature (control) or simulated supply chain conditions were significantly firmer (3.0 and 19.1% reduction in firmness values from harvest, respectively) compared to the counterpart fruit from the conventional treatment (17.4 and 34.4% reduction in firmness values from harvest, respectively) (Figure 2B). 

### 3.3. Weight Loss

As expected, weight loss increased during storage, regardless of the treatment (Figure 3). Weight loss in ‘Florida Radiance’ strawberries from the first harvest was, on average, lower (6.3%) than in fruit from the second harvest (8.0%), which may have contributed to more distinct differences between treatments regarding sensory quality, color, texture, and most chemical components (Figure 3A). In the first harvest, weight loss was higher in fruit from the conventional supply chain treatment (6.6%) than in fruit from the reduced control treatment (6.0%). Still, there was no significant difference (*p* > 0.05) in the weight loss of strawberries from the conventional control treatment (6.3%) and the reduced supply chain fruit (6.3%). In the second harvest, strawberries from the conventional and reduced supply chain treatments had the highest weight loss (8.0 and 8.3%, respectively). In contrast, fruit from the reduced control treatment had the lowest weight loss (7.5%) than the other treatments (Figure 3B). 

### 3.4. Acidity and Soluble Solids Content

The acidity of ‘Florida Radiance’ strawberries decreased significantly with time, regardless of the harvest or the treatment (Figure 4A,B). However, the decrease in acidity levels was more substantial in fruit from the second harvest than the first harvest (60.6% and 45.3%, respectively), explaining the differences between treatments obtained for the two harvests. In the first harvest, the acidity of conventional and reduced fungicide strawberries from the control treatment decreased by 44.0% and 43.1%, respectively. In contrast, the acidity of conventional and reduced fungicide fruit from the supply chain treatment decreased by 46.7% and 47.3%, respectively (Figure 4A). In the second harvest, the acidity of conventional and reduced fungicide strawberries from the control treatment was reduced by 65.1% and 57.7%, respectively. The acidity of conventional and reduced fungicide fruit from the supply chain treatment was decreased by 59.4% and 60.5%, respectively (Figure 4B). 

The soluble solids content (SSC) of ‘Florida Radiance’ strawberries also declined significantly (*p* < 0.05) during storage regardless of harvest or treatment (Figure 4C,D). However, the decline was greater in fruit from the second harvest than from the first harvest (49.4% and 57.6%, respectively). Overall, the decrease in the SSC was higher in strawberries exposed to supply chain conditions than in a steady temperature (control). In the first harvest, conventional and reduced fungicide fruit from the control treatments had a similar decline in the SSC (47.3% and 47.9%, respectively) as well as conventional and reduced fungicide fruit from the supply chain treatments (51.0% and 51.4%, respectively) (Figure 4C). In the second harvest, the trend was similar. Conventional and reduced fungicide fruit from the control treatments had a similar decrease in SSC (45.8% and 55.0%, respectively) as well as conventional and reduced fungicide fruit from the supply chain treatments (57.4% and 52.3%, respectively) (Figure 4D). 

### 3.5. Bioactive Compounds

The levels of total phenolics (TPC), anthocyanins (ANC), and ascorbic acid (AA), significantly decreased (*p* < 0.05) during storage regardless of the date of harvest or temperature treatment (Figure 5). However, such as that observed for acidity and SSC, the decline in bioactive components was less in ‘Florida Radiance’ strawberries from the first harvest than from the second harvest, most likely due to the higher weight loss in fruit from the second harvest (Figure 3). 

The decline in TPC in fruit from the conventional control treatment was 47.5% and 54.8%, whereas in fruit from the conventional supply chain TPC declined 52.5% and 60.3% for the first and second harvest, respectively (Figure 5A,B). Strawberries from the reduced control treatment also declined less in TPC (43.5% and 48.0%, first and second harvest, respectively) than in the fruit from the reduced supply chain treatment (45.1% and 51.8%, respectively).

After the supply chain simulation, ANC was higher in reduced fungicide strawberries from the first harvest than fruit from the conventional treatment (Figure 5C). Even though at harvest, conventional strawberries had higher ANC than fruit from the reduced fungicide treatment (1.68 and 1.34 g kg^−1^, respectively), the decline in anthocyanin levels was lower in reduced fungicide fruit. In contrast, in the second harvest, the trend was slightly different (Figure 5D). Similarly to the first harvest, fruit from the conventional treatment had, at harvest, a higher ANC than the strawberries from the reduced fungicide treatment (1.46 and 1.30 g kg^−1^, respectively). However, the ANC decline was higher for fruit from the control reduced fungicide treatment than for the control conventional treatment (36.9% vs. 19.2%, respectively). Regardless, after 6 days, strawberries from the reduced supply chain simulation lost less ANC than the conventional supply chain fruit (28.5% vs. 32.2%, respectively). 

In the first harvest, AA content was lower in ‘Florida Radiance’ strawberries from the conventional treatment compared to the reduced fungicide fruit (3.9% and 4.9%, respectively). After 6 days, conventional strawberries also had a lower AA than the reduced fungicide fruit (Figure 5E,F). The decline in AA was lower in strawberries from the reduced control and reduced supply chain treatments (28.9% and 34.7%, respectively) than in fruit from the conventional treatment (40.6% decline in conventional control, and 38.0% decline in the conventional supply chain). In the second harvest, the trend was similar, with strawberries from the reduced fungicide treatment showing higher AA content after 6 days than conventional fruit. 

### 3.6. Sugar Profiles

At harvest, sucrose content was higher in ‘Florida Radiance’ strawberries from the reduced fungicide treatment than in conventional fruit (Figure 6A,B). Overall, after 6 days, fruit from the control treatments tended to have higher sucrose levels than fruit exposed to supply chain conditions. In the first harvest, the most significant decline in sucrose levels was observed in reduced fungicide strawberries from the control treatment (74.2% decline). In contrast, there was no significant difference between the other treatments (Figure 6A). In the second harvest, conventional strawberries from the supply chain treatment showed the highest decrease in sucrose content (83.4%) after 6 days. In comparison, reduced fungicide fruit from the control treatment showed the lowest decrease (57.4%) and a higher sucrose content than the remaining treatments (Figure 6B). 

Although the trend was similar, glucose and fructose also decreased but at a lower rate than sucrose. In the first harvest, reduced fungicide strawberries from the control treatment had the highest glucose levels and the most negligible decline after 6 days (39.4%) than the other treatments (Figure 6C). Conventional strawberries exposed to supply chain conditions had the highest drop in glucose levels (49.5%) compared to reduced fungicide fruit exposed to the same conditions (45.4%). In the second harvest, although reduced fungicide strawberries from the control treatment had a slower decline in glucose levels than the other treatments after 6 days, differences between treatments were not significant (Figure 6D). As for fructose, reduced fungicide strawberries from the control treatment showed the least decline (36.5% and 44.2% for the first and second harvests, respectively) compared to conventional strawberries exposed to supply chain conditions that showed the most significant decline in the fructose content (45.5% and 52.2% for the first and second harvest) (Figure 6E,F).

### 3.7. Residual Fungicides

In the first harvest, at the end of the supply chain simulation (i.e., at the consumer level), analysis of residual fungicides showed that unwashed conventional ‘Florida Radiance’ strawberries exposed to supply chain conditions had significantly higher levels of fludioxonil, cyprodinil, pyraclostrobin (only detected in supply chain unwashed fruit) and captan compared to washed fruit exposed to the same conditions (Table 4). Compared to the control group, strawberries exposed to supply chain conditions (either washed or unwashed) had significantly higher levels of fungicides (*p* < 0.05). Strawberries from the reduced fungicide treatment had, on average, substantially lower fungicide levels than conventional fruit, but the differences within the reduced fungicide treatments were less pronounced. Thus, the highest fludioxonil and cyprodinil levels were found in unwashed reduced fungicide strawberries from the control treatment. In contrast, the highest levels of captan were found in unwashed fruit from the supply chain treatment. Washed strawberries from the reduced pesticide treatment, either from the control or supply chain, had lower captan levels than unwashed fruit. In the second harvest, results were slightly different compared to those from the first harvest. The highest levels of fludioxonil, cyprodinil, and captan were detected in unwashed conventional strawberries from the control treatment. In contrast, the lowest levels of fungicides were detected in washed fruit exposed to supply chain conditions. In strawberries from the reduced fungicide treatment, fludioxonil and captan were also lower in fruit exposed to supply chain conditions. In contrast, washed fruit from the control treatment had lower levels of cyprodinil.

### 3.8. Microbial Load

Microbial load on ‘Florida Radiance’ strawberries was evaluated at harvest, after storage at the distribution center (DC), and at the consumer level on unwashed and washed strawberries. In the first harvest, at the time of harvest (0 h), strawberries from the reduced fungicide treatment had significantly lower CFU g^−1^ (*p* < 0.05) compared to conventionally grown fruit (4.12 and 4.52 log CFU g^−1^, respectively) (Figure 7A). Microbial populations tended to increase during storage, and at the DC (4 days after harvest), the highest microbial counts were recorded in conventional strawberries kept at a constant temperature, followed by reduced fungicide fruit exposed to supply chain conditions. Strawberries from the reduced fungicide treatment kept at a constant temperature had the lowest microbial counts among treatments. At the consumer level (6 days after harvest), the microbial load increased in unwashed strawberries but was higher in conventional strawberries than reduced fungicide fruit. Washing the fruit reduced the microbial populations in conventional fruit, but there was no consistent result with reduced fungicide fruit. In the second harvest, results were similar; at the time of harvest (0 h), strawberries from the reduced fungicide treatment had a significantly lower CFU g^−1^ compared to conventionally grown fruit (5.17 and 5.52 log CFU g^−1^, respectively) (Figure 7B). After storage at the DC and at the consumer level, unwashed conventional strawberries exposed to supply chain conditions had significantly higher microbial loads. Washing the fruit significantly reduced (*p* < 0.05) the microbial populations in conventional strawberries from the control treatment. 

## 4. Discussion

Environmental concerns, pesticide contamination, and consumer awareness have contributed to the increasing demand for better food quality. A clearer understanding of food wasted along the supply chain and issues related to food security have also become global concerns. Hence, there is a need to produce more food with a better overall quality while reducing waste from the field to the consumer. 

Fresh fruits and vegetables, and strawberries particularly, are most often contaminated by residual pesticides and wasted along the supply chain. Thus, several studies have compared the quality of strawberries grown under conventional and organic agricultural conditions [40,41,42,43,44,45,46]. Others have also reported on the levels of pesticide residues in organic, integrated pest management (IPM), and conventionally grown strawberries [40,41,42,43,44,45,46]. However, there is a lack of studies that report the impact of supply chain conditions on the overall quality of strawberries grown under various agricultural practices. To our knowledge, only one study has been published on the effect of using reduced pesticide applications compared to conventional and organic methods on strawberry quality [15]. In the study, the authors compared the overall quality and residual fungicide levels of two strawberry cultivars, ‘Strawberry Festival’ and ‘Florida Radiance’, grown using repeated (i.e., conventional), reduced fungicide or non-fungicide (i.e., organic) applications at harvest and during a steady storage temperature for 7 days at 1.5 °C [15]. Strawberries grown under reduced fungicide applications have a similar or better physicochemical quality than conventionally and organically grown fruit but lower levels of fungicide residues than conventional fruit [15]. In a separate study, the impact of each step along the supply chain on the quality of conventionally grown ‘Florida Radiance’ strawberries was evaluated. We showed that supply chain conditions have a negative effect on the overall quality of strawberries, particularly when the fruit is exposed to non-optimal temperature conditions such as storage at the grower at 5 °C, shipping to the stores at 8 °C, and storage at the consumer level at 20 °C [7]. In the present study, we combined the two approaches used in our previous research, pre-harvest (i.e., disease control treatments) and postharvest factors (i.e., optimum steady temperature and supply chain conditions), to investigate the impact of supply chain conditions on ‘Florida Radiance’ strawberries grown under different disease control treatments. Strawberries were grown under conventional and reduced fungicide applications, exposed to optimum constant temperature or simulated supply chain conditions from the farm to the consumer, and the fruit was analyzed for sensory and physicochemical quality as well as for residual fungicides and microbial load before and after washing the fruit.

### 4.1. Effect of Cultivation Methods and Supply Chain Conditions on Sensory Quality, Color, and Texture

Disease control treatments (i.e., conventional or reduced fungicide) used in this study had a less significant effect on the sensory quality of ‘Florida Radiance’ strawberries than the effect of supply chain conditions (i.e., constant or high fluctuating temperatures) had on sensory quality. However, strawberries kept at a constant temperature maintained a better overall sensory quality than those exposed to supply chain conditions, regardless of the disease control treatment. Several steps along the supply chain may impact the appearance of strawberries [1,6,7,9]. For example, in a study using “real-life” supply chain conditions, the appearance of ‘Albion’ strawberries exposed to supply chain conditions deteriorated considerably compared to fruit kept at a constant temperature. Supply chain strawberries appeared darker, overripe, and the calyxes were dry and wilted upon arrival to the distribution center (DC), while fruit kept at a constant temperature had a better-quality appearance [6]. In a study using simulated supply chain conditions, high temperatures during pre-cooling and storage at the grower, shipping to and storage at the DC, shipping to the supermarket, and display at the store were identified as the steps along the supply chain that most impacted the ‘Florida Radiance’ strawberry’s quality appearance [7].

There is minimal information regarding consumer acceptability towards organic, IPM, or reduced fungicide vs. conventionally grown strawberries. In a previous study, we have shown that the appearance of ‘Florida Radiance’ and ‘Strawberry Festival’ conventional strawberries was not significantly different from reduced fungicide or organic fruit [15]. Others suggested that even though the redder color of some organic strawberry cultivars (‘Diamante’, ‘Lanai’ and ‘San Juan’) might be preferred over their conventional counterparts, consumer preference between conventional and organic strawberries seems to be cultivar-dependent and not influenced by the type of cultivation method [21]. In another study, higher external color scores were given to organically grown ‘Selva’ strawberries due to their redder color compared to conventionally grown fruit [18]. In our study, the appearance of ‘Florida Radiance’ strawberry from the reduced control treatment obtained higher scores but not always significantly different from the counterpart conventional control treatment (Table 3). Thus, without further consumer studies, it is not accurate to conclude that the sensory quality of organic or reduced pesticide fruit is superior to that of strawberries conventionally grown. 

Strawberry surface color measurements recorded using a CIE-L*a*b* uniform color space showed that strawberries from the reduced fungicide treatment tended to be darker red (lower L* and hue values) than conventional fruit (Figure 1). We found no significant difference in strawberry surface color between cultivars grown under conventional and reduced fungicide treatments [15]. However, others have reported that the color of organic strawberries tends to be darker red (lower L* and hue values) compared to conventionally grown fruit of the same cultivar [18,21]. Differences in color between cultivation methods have been associated with higher levels of anthocyanins in organic, IPM, and reduced fungicide than conventionally grown strawberries [15,18,42].

Softening of strawberry fruit has been associated with increased storage temperatures, leading to increased moisture loss and decreased cell turgidity [47]. Supply chain conditions used in the present study led to fruit softening, particularly in conventionally grown strawberries from the second harvest. ‘Florida Radiance’ strawberries from the reduced control treatment were significantly firmer than fruit from other treatments (Figure 2). Similarly, in a previous study, we also found that strawberry cultivars from reduced fungicide treatments softened less during storage than fruit from the conventional treatment, while strawberries exposed to simulated supply chain conditions were softer than fruit kept at constant temperatures [7,15].

### 4.2. Effect of Cultivation Methods and Supply Chain Conditions on Chemical Attributes

‘Florida Radiance’ strawberries from the first harvest had, on average, lower weight loss than fruit from the second harvest (Figure 3). Moreover, the reduced fungicide control treatment, harvested later in the season, had a significantly lower weight loss than conventional control strawberries. Although to our knowledge, there are no published studies that report differences between weight loss of strawberries grown using conventional and reduced fungicide agricultural methods, previous studies have reported a lower weight loss in organic compared to conventionally grown strawberries [21]. These authors suggested that slower dehydration in organic fruit may be due to a thicker cuticle and expansion of epidermal cell walls. Studies have also shown that strawberries exposed to supply chain conditions lose more weight than fruit kept at a constant temperature [7,9,11]. In the present study, the negative impact of supply chain conditions on the weight loss of ‘Florida Radiance’ strawberries was more evident in the second harvest and in fruit from the reduced fungicide treatment. 

Due to the difference in weight loss between harvest and treatments, chemical attributes were expressed as dry weight. Although it might be argued that chemical, bioactive, and sugar contents expressed on a fresh weight basis represent the actual concentrations that consumers would experience, expressing the data on a dry weight basis shows the actual differences between harvest and treatments that might have been obscured due do water loss. In general, the higher the weight loss, the more significant the decline in strawberry chemical attributes [6,7,36]. Water loss results in a reduced turgor pressure of the cells leading to membrane breakdown. For example, the collapse of cellular membranes allows for degrading enzymes, such as polyphenol oxidase (PPO), to act on polyphenols resulting in the formation of brown pigments and a decrease in the total polyphenol content [37,48]. Therefore, in this study, strawberries from the first harvest had, on average, a lower weight loss (6.3%), regardless of the treatment, compared to fruit from the second harvest (8.0%) and, as a result, had on average a smaller decline in the levels of chemical, bioactive and sugar contents.

Conventional and reduced pesticide fruit exposed to supply chain conditions had a greater decrease in acidity compared to the control. In a previous study, we found the same trend with strawberries stored at the DC at 5 °C or shipped to stores at 8 °C having lower acidity levels than fruit maintained at an optimum constant temperature [7]. Looking at the differences in acidity levels between harvests and treatments, strawberries from the reduced supply chain treatment from the first harvest had lower acidity levels than conventional fruit at harvest and after 6 days. In contrast, in the second harvest, conventional fruit had the lowest acidity levels at harvest and after 6 days than fruit from the reduced fungicide treatments. Slight differences in the maturity of the fruit at harvest and loss of moisture during storage can influence the levels of acids in strawberries which may explain the differences between harvests [7]. Fruit from the first harvest lost less weight than that from the second harvest, which may have also impacted the composition of the fruit from the different treatments. In previous studies, we also showed that trends in acidity levels are cultivar-dependent and are also influenced by the maturity of the fruit at harvest. For example, there was no significant difference in the acidity of ‘Strawberry Festival’ strawberries from different cultivation methods, but ‘Florida Radiance’ acidity levels from the reduced pesticide treatment were higher than the conventional treatment [15]. Other studies reported no consistent trend in the acidity of organic vs. conventional strawberries. Organic strawberries from different cultivars had higher, similar or lower acidity than their conventional counterparts [21,41].

Overall, in the first harvest, ‘Florida Radiance’ conventional and reduced fungicide strawberries exposed to supply chain conditions had a lower SSC than control fruit. In the second harvest, the trend was similar, but strawberries from the control reduced pesticide treatment had, after 6 days, a higher SSC than the other treatments. Previous studies also showed a decline in the SSC of strawberries exposed to supply chain conditions [6,7]. Moreover, after 7 days of storage at 1.5 °C, the SSC of ‘Strawberry Festival’ conventional strawberries was significantly lower than fruit from reduced pesticide treatment, but there was no significant difference in the SSC of ‘Florida Radiance’ from both treatments [15]. In another study, ‘Diamante’ organic strawberries had a higher SSC than their conventional counterparts; however, there were no significant differences between organic and conventional ‘Lanai’ and ‘San Juan’ strawberries [21]. Others have shown no significant difference in the SSC between organic and conventional strawberries [41]. Thus, although reduced pesticide strawberries tended to have higher SSC than conventional fruit, cultivar variability and time–temperature regimes during the supply chain seem to have a more significant impact on acidity and the SSC than the disease control treatment applied in the field.

### 4.3. Effect of Cultivation Methods and Supply Chain Conditions on Bioactive Compounds

Strawberry bioactive compounds, namely ascorbic acid (AA), total phenolics (TPC), and anthocyanins (ANC), have been the subject of many studies due to their proven beneficial impact on human health [49,50,51,52,53,54]. However, agricultural practices combined with adverse supply chain conditions may significantly impact the levels of bioactive compounds in strawberries. Although studies comparing organic against conventional strawberries are somehow controversial, it is now well established that biotic and abiotic stress triggers the plant defense system, leading to it synthesizing a large diversity of polyphenols [42,55,56,57,58]. However, little is known about the impact of pre-harvest factors and postharvest temperature or supply chain conditions on the synthesis and degradation of strawberry bioactive compounds.

In our study, the decreases in AA and TPC in ‘Florida Radiance’ strawberries from the first harvest were approximately 47.2% and 40.6%, respectively, whereas, in the second harvest, there was a 53.7% and 43.3% decline in these components. Conversely, the ANC decline was higher in the first harvest (40.6%) than in the second harvest (29.2%). Because all chemical components were expressed in terms of dry weight, the effect of concentration due to water loss during storage does not explain such difference as strawberries from the second harvest lost on average more water (weight) than those from the first harvest. One possible explanation is that in fruit from the second harvest, higher water loss resulted in less available free water in the tissues, contributing to a slower rate of anthocyanin degradation. Nonetheless, the cultivation method and supply chain conditions affected the levels of bioactive compounds in ‘Florida Radiance’ strawberries. A previous study also found that supply chain conditions, particularly storage at the grower and shipping to the distribution center and retail stores at non-optimal temperatures, contributed to a significant decline (up to 57.0%) in bioactive compounds [7]. In contrast, after cold storage, strawberries from the reduced pesticide treatment had a greater or similar anthocyanin content than the conventional disease control treatment [15,22]. Several other studies have also shown that at harvest, strawberries from IPM or organic farming had similar or higher amounts of vitamin C, phenolic compounds, and anthocyanins than conventionally grown fruit [17,18,19,21,23,40,41,42]. Overall, in our study, strawberries from control treatments showed less decline in bioactive compounds than fruit exposed to supply chain conditions, regardless of the cultivation method. In contrast, reduced fungicide fruit had, on average, higher levels of bioactive compounds compared to their conventional counterparts.

### 4.4. Effect of Cultivation Methods and Supply Chain Conditions on Sugar Profiles

Sucrose, glucose, and fructose, the main sugars found in strawberries, decreased during storage at both a constant temperature and supply chain conditions, regardless of the treatments. In general, sucrose contributes the least to the total sugar content in strawberries, whereas glucose and fructose concentrations are at about a 1:1 ratio [59]. In our study, conventional ‘Florida Radiance’ strawberries had, on average at harvest, lower sucrose but higher glucose and fructose contents compared to reduced fungicide fruit. Moreover, in a previous study, organic strawberries had a lower sucrose content at harvest but more elevated glucose and fructose levels than conventional and reduced fungicide strawberries [15]. Some have reported no difference in the reducing (glucose and fructose) or total sugar content between conventional and organic strawberries [21]. Yet others have suggested that the lower levels of sugars in organic fruit may result from the lower amounts of fertilizer used in organic than conventional production [56]. Such agricultural practices boost vegetative growth resulting in increased synthesis of primary nutrients such as carbohydrates and lower synthesis of secondary metabolites such as polyphenols [56]. Even though the sugar content in conventional strawberries may be high at harvest, results from our study suggest that supply chain conditions may contribute to a greater decline in the sugar content than an optimum constant temperature. A previous study also found that the total sugar content was negatively impacted by supply chain conditions, particularly during storage at non-optimum temperatures at the distribution center and the consumer [7]. A decrease in sugar contents results from the normal metabolic activity after the fruit being detached from the mother plant. The increase in the overall rate of metabolic reactions (e.g., glycolysis) with increased time–temperature exposure contributes to the breakdown of sugars, depleting fruit reserves. Regardless, it seems that strawberries from the reduced fungicide treatment tended to have, on average, a higher sugar content after exposure to supply chain conditions compared to conventional fruit. In a previous study, we found that at harvest ‘Florida Radiance’ strawberries from the reduced fungicide treatment had higher glucose and fructose contents than conventional fruit, but there was no difference in the total sugar content after cold storage [22]. 

### 4.5. Effect of Cultivation Methods and Supply Chain Conditions on Residual Fungicides

Fungicides are commonly used in conventional agriculture to protect strawberries from field rots and improve yield. Studies have found that up to 93.7% of the agricultural land in Europe, North America, and South America is contaminated by more than one active ingredient typically found in pesticide mixtures [60]. Although organic farming can produce pesticide-free strawberries, the yield might be lower than conventional, or the cost associated with organic production might be too high for some world regions. Thus, there is a need to reduce pesticide applications while maintaining yield, reducing costs, and meeting the global food demand. Among the crop protection chemicals used on strawberries, fungicides used to control diseases correspond to the majority (≈40%) of the chemical applications. We have developed and shown that by utilizing a disease forecasting system, the application of fungicides can be reduced by 50% or more while achieving the same level of disease control and without affecting yields of strawberry quality [15,24,26]. 

The United States Environmental Protection Agency (USEPA) established tolerances of 2000 ppb for captan, 3000 ppb for fludioxonil and penthiopyrad, 5000 ppb for cyprodinil, 200 ppb for cyflufenamid, 10,000 ppb for azoxystrobin, and 1200 ppb for pyraclostrobin [61]. Therefore, the levels of fungicides found in strawberries from any disease control treatments used in this study were well below the tolerances established by the USEPA. A previous study comparing conventional to reduced fungicide strawberries also showed that, although the levels of fungicides were higher in conventional fruit, they were well below the limits established by USEPA [15]. Other studies have shown that the levels of pesticides in strawberries from IPM agricultural systems fall in between conventional and organic productions [45]. In the present study, ‘Florida Radiance’ strawberries from the reduced fungicide treatment had significantly lower fungicide residues than fruit from conventional treatment. On average, the total level of fungicide residues in strawberries exposed to supply chain treatments was higher than fruit maintained at a constant temperature (4100 and 3790 ppb, respectively). Although no studies have shown how levels of fungicides may change during storage at a constant temperature or during supply chain conditions, it is possibly such as that observed after field applications [62,63,64]. Residues dissipate differently with half-lives under different time–temperature postharvest scenarios. Nonetheless, washing the fruit at the consumer level significantly reduced fungicide residues compared to unwashed fruit. Thus, regardless of the disease control treatment, or the time–temperature scenario after harvest, washing the fruit at the consumer level significantly reduced fungicides on strawberries. Other studies have also reported that all pesticide residues on strawberries were reduced up to 67.8% after rinsing the fruit [64,65]. Overall, the benefits of reducing human exposure to pesticides by substituting conventional organic produce might be negligible because the exposure to pesticide residues from consuming a standard diet poses minimal risk to humans [66].

### 4.6. Effect of Cultivation Methods and Supply Chain Conditions on Microbial Load

*Botrytis cinerea* and *Rhizopus stolonifer* are the main fungal pathogens found in strawberries during postharvest handling. Higher temperatures during the strawberry supply chain contribute to an increased fungal development causing spoilage, particularly at the retail and consumer levels [8,16]. Moreover, in recent years, concerns regarding the safety of strawberries have increased due to some outbreaks of foodborne pathogens linked to consumption [13]. However, very little data are available comparing microbial and fungal contamination of strawberries grown under different disease control treatments at harvest or after exposure to supply chain conditions [12,14,67,68]. For example, Flessa et al. [12] showed that survival of pathogens on strawberry surfaces generally decreases when storage temperatures are maintained below 10 °C. In another study, we showed that, although there was no visible decay during an 8-day storage period, aerobic plate counts and yeast and molds were significantly higher in strawberries stored at 10 °C than in fruit stored at 1 °C [68]. 

In our study, ‘Florida Radiance’ strawberries exposed to constant or supply chain conditions did not show a conclusive microbial load trend. However, in the first harvest, non-washed strawberries from the reduced fungicide treatment had significantly lower microbial loads than conventional fruit at the consumer level. Results from our study also showed that at harvest, conventional strawberries tended to have higher microbial loads than fruit from the reduced fungicide treatment. One study showed that the microbial composition of freshly harvested strawberries was more complex than that of fruit sampled at the market, suggesting that microbial loads and composition on strawberry surfaces is influenced by soil, irrigation water and that some microbiota may disappear when the fruit arrives at the market or after consumer washing [14]. Another study concluded insignificant differences between the microbiota on strawberries from conventional agriculture that received up to 10 fungicide spray treatments and organic strawberries [67]. Nonetheless, our study showed that washing the fruit at the consumer level resulted in a 6% reduction in microbial counts despite the disease control treatment used.

## 5. Conclusions

This study showed that ‘Florida Radiance’ strawberries grown under reduced fungicide applications had a better or similar overall quality than their counterparts grown under conventional methods, but they also had much lower residual fungicides—fludioxonil, cyprodinil, pyraclostrobin, and captan. Although the levels of residual fungicides on strawberries from either disease control treatment were below the tolerances established, our results validate the importance of washing strawberries before consumption to reduce fungicide and microbial loads (6% reduction) on the fruit. Moreover, regardless of the disease control treatment used, the overall quality of ‘Florida Radiance’ strawberries was better maintained when the fruit was handled at a constant optimum temperature throughout the supply chain, from the field to the consumer. Overall, disease control systems that use a lower number of fungicide sprays during the strawberry production season are a promising alternative to conventional agricultural practices. Thus, using reduced fungicide systems combined with proper handling during the supply chain helps reduce strawberry waste while providing the consumer fruit with better quality. Further studies using various strawberry cultivars grown in different regions of the world would give a deeper understanding of the importance of agricultural and supply chain practices on the overall quality of strawberries.

## Figures and Tables

**Figure 1 foods-10-01442-f001:**
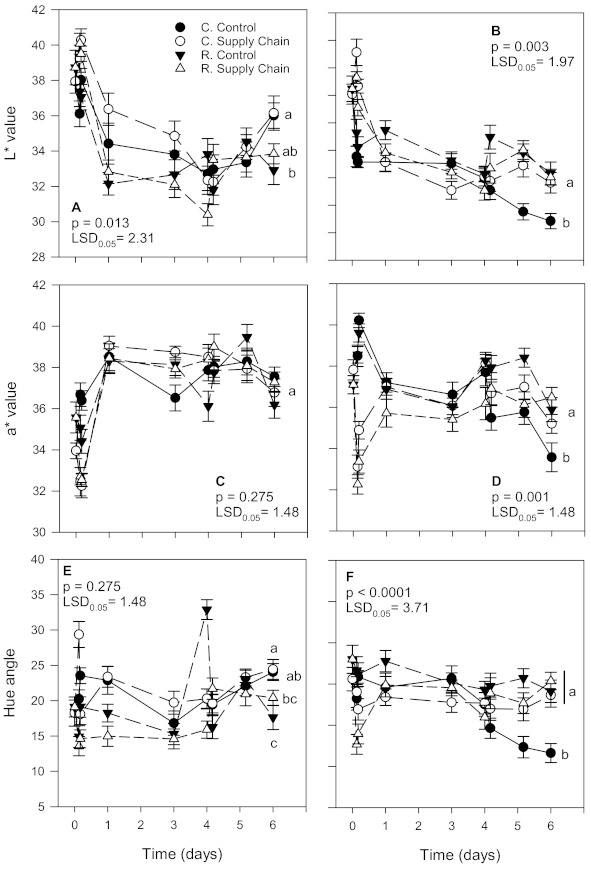
L*, a *, and hue angle values of ‘Florida Radiance’ strawberries from conventional (C) and reduced fungicide (R) disease control treatments held under a steady temperature regime at 1.5 °C and 85% RH (control) or exposed to simulated supply chain conditions. First harvest (**A**,**C**,**E**) and second harvest (**B**,**D**,**F**). Data points are means of 3 biological replicates of 12 strawberries each. Different letters indicate significant differences (*p* ≤ 0.05) between treatments after 6 days at a constant temperature or simulated supply chain conditions.

**Figure 2 foods-10-01442-f002:**
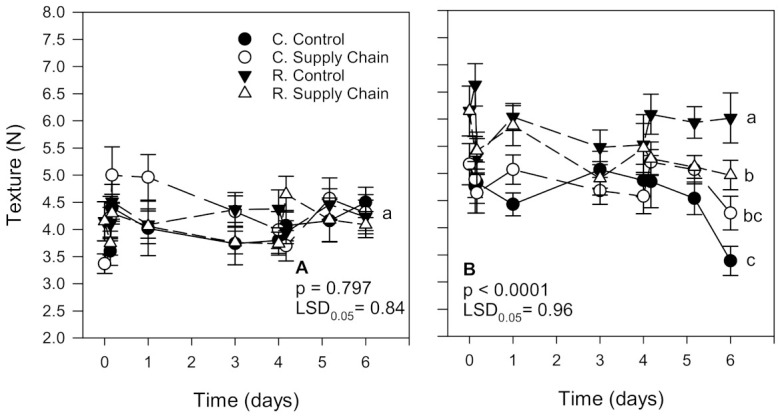
Texture of ‘Florida Radiance’ strawberries from conventional (C) and reduced fungicide (R) disease control treatments held under a steady temperature regime at 1.5 °C and 85% RH (control) or exposed to simulated supply chain conditions. First harvest (**A**) and second harvest (**B**). Data points are means of 3 biological replicates of 12 strawberries each. Different letters indicate significant differences (*p* ≤ 0.05) between treatments after 6 days at a constant temperature or simulated supply chain conditions.

**Figure 3 foods-10-01442-f003:**
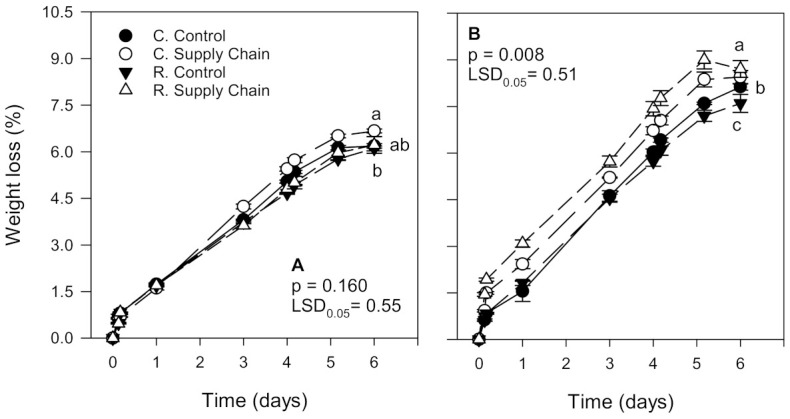
Weight loss of ‘Florida Radiance’ strawberries from conventional (C) and reduced fungicide (R) disease control treatments held under a steady temperature regime at 1.5 °C and 85% RH (control) or exposed to simulated supply chain conditions. First harvest (**A**) and second harvest (**B**). Data points are means of 3 biological replicates of 12 strawberries each. Different letters indicate significant differences (*p* ≤ 0.05) between treatments after 6 days at a constant temperature or simulated supply chain conditions.

**Figure 4 foods-10-01442-f004:**
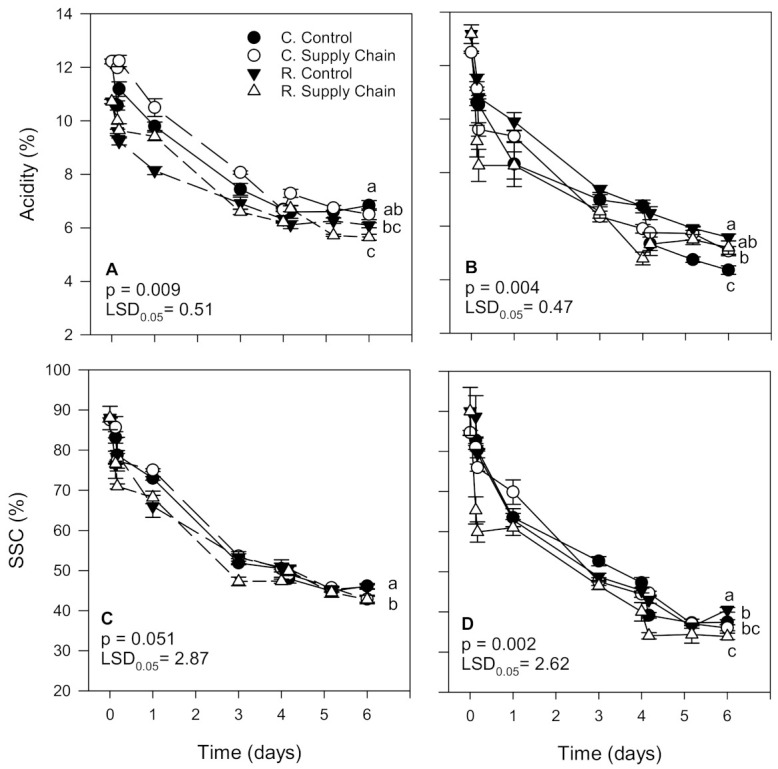
Acidity and soluble solids content (SSC) of ‘Florida Radiance’ strawberries from conventional (C) and reduced fungicide (R) disease control treatments held under a steady temperature regime at 1.5 °C and 85% RH (control) or exposed to simulated supply chain conditions. First harvest (**A**,**C**) and second harvest (**B**,**D**). Data points are means of 3 biological replicates of 12 strawberries each. Different letters indicate significant differences (*p* ≤ 0.05) between treatments after 6 days at a constant temperature or simulated supply chain conditions.

**Figure 5 foods-10-01442-f005:**
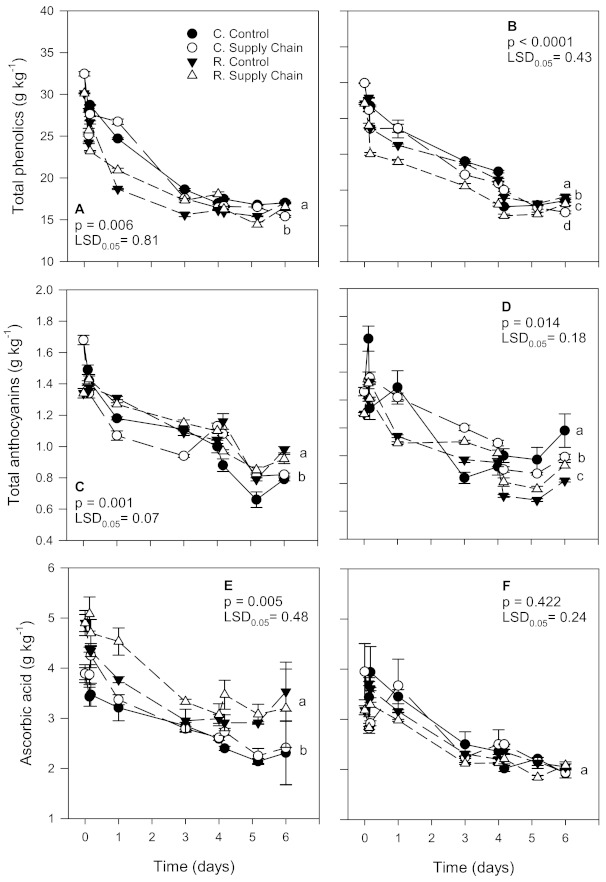
Levels of bioactive compounds (total phenolics, anthocyanins, and ascorbic acid) of ‘Florida Radiance’ strawberries from conventional (C) and reduced fungicide (R) disease control treatments held under a steady temperature regime at 1.5 °C and 85% RH (control) or exposed to simulated supply chain conditions. First harvest (**A**,**C**,**E**) and second harvest (**B**,**D**,**F**). Data points are means of 3 biological replicates of 12 strawberries each. Different letters indicate significant differences (*p* ≤ 0.05) between treatments after 6 days at a constant temperature or simulated supply chain conditions.

**Figure 6 foods-10-01442-f006:**
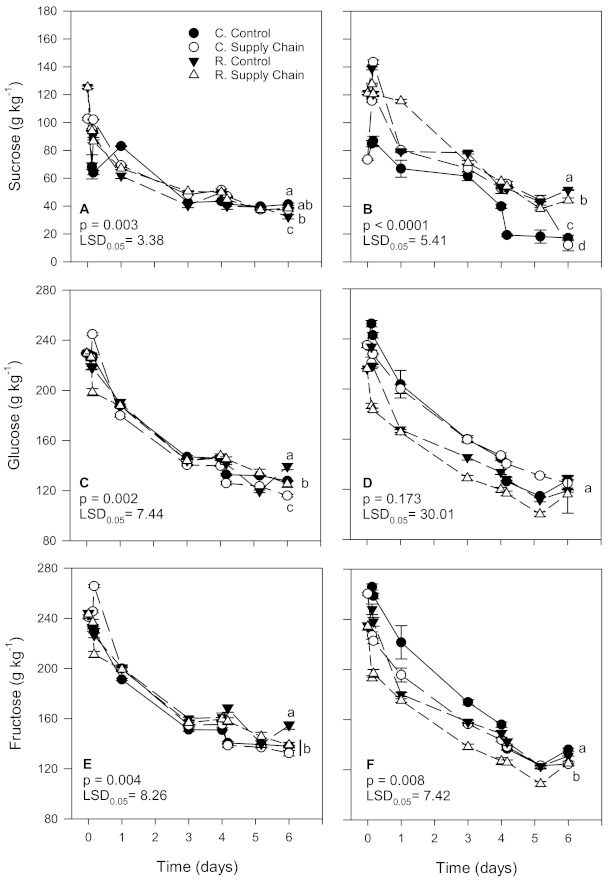
Sucrose, glucose, and fructose contents of ‘Florida Radiance’ strawberries from conventional (C) and reduced fungicide (R) disease control treatments held under a steady temperature regime at 1.5 °C and 85% RH (control) or exposed to simulated supply chain conditions. First harvest (**A**,**C**,**E**) and second harvest (**B**,**D**,**F**). Data points are means of 3 biological replicates of 12 strawberries each. Different letters indicate significant differences (*p* ≤ 0.05) between treatments after 6 days at a constant temperature or simulated supply chain conditions.

**Figure 7 foods-10-01442-f007:**
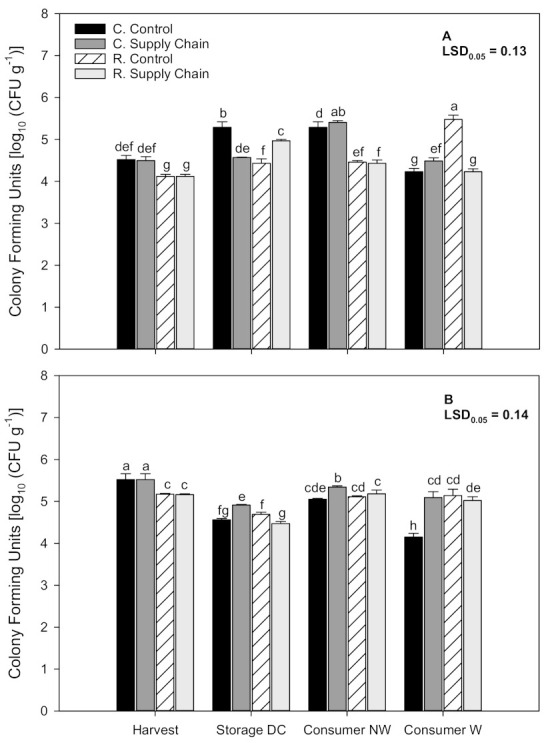
Microbial load in ‘Florida Radiance’ strawberries from conventional (C) and reduced fungicide (R) disease control treatments held under a steady temperature regime at 1.5 °C and 85% RH (control) or exposed to simulated supply chain conditions. Evaluations were conducted at harvest and after storage at the distribution center (DC) on non-washed fruit and at the consumer level on non-washed (NW) and washed (W) strawberries. First harvest (**A**) and second harvest (**B**). Data points are means of 3 biological replicates of 12 strawberries each. Different letters indicate significant differences (*p* ≤ 0.05) between treatments after 6 days at a constant temperature or simulated supply chain conditions.

**Table 1 foods-10-01442-t001:** Fungicides applied to ‘Florida Radiance’ grown in Plant City, FL, USA, under conventional and reduced fungicide disease control treatments.

Date	Conventional	Reduced
13 January 2015	Captan + Thiram (a.i. ^1^ captan, thiram)	Captan + Thiram (a.i. captan, thiram)
17 January 2015	Fontelis (a.i. penthiopyrad)	Fontelis (a.i. penthiopyrad)
22 January 2015	Captan + Thiram (a.i. captan, thiram)	
24 January 2015	Merivon (a.i. fluxapyroxad, pyraclostrobin)	
28 January 2015	Captan + Thiram (a.i. captan, thiram)	
4 February 2015	Captan + Thiram (a.i. captan, thiram)	
9 February 2015	Captan + Thiram (a.i. captan, thiram)	Captan + Thiram (a.i. captan, thiram)
12 February 2015	Switch (a.i. cyprodinil, fludioxonil)	
22 February 2015	Captan + Thiram (a.i. captan, thiram)	
3 March 2015	Harvest 1	
9 March 2015	Captan (a.i. captan)	Captan (a.i. captan)
10 March 2015	Harvest 2	
14 March 2015	Captan (a.i. captan)	

^1^ a.i—active ingredient.

**Table 2 foods-10-01442-t002:** Strawberry supply chain simulation from the field to the consumer ^1^.

		Control		Supply Chain	
Supply Chain Steps	Duration (Days)	Temperature (°C)	RH (%)	Temperature (°C)	RH (%)
Harvest	0.00	-	-	-	-
Harvest to pre-cooling ^2^	0.13	1.5	85.0	30.0	60.0
Pre-cooling	0.04 (0.17) ^3^	1.5	85.0	2.0	70.0
Cold room (grower)	0.83 (1.00)	1.5	85.0	1.5	80.0
Shipping to D.C ^4^	2.00 (3.00)	1.5	85.0	3.0	85.0
Storage at D.C	1.00 (4.00)	1.5	85.0	4.0	80.0
Transport from D.C. to store	0.17 (4.17)	1.5	85.0	5.0	83.0
Store display	1.00 (5.17)	1.5	85.0	7.0	80.0
Consumer ^5^	0.83 (6.00)	1.5	85.0	4.0	30.0
Total Duration	6 days				

^1^ Temperature and RH are average values calculated based on data previously collected [1,6,29]; ^2^ Delays before pre-cooling, waiting in the field; ^3^ Cumulative days are given in parenthesis; ^4^ D.C.—Distribution center in the US Midwestern States or Eastern Canada; ^5^ Sensory analysis was conducted at the consumer level after washing the fruit; residual pesticide analysis was conducted at the consumer level before and after washing the fruit; microbial analysis was performed at harvest, after storage at the DC, and consumer level before and after washing the fruit.

**Table 3 foods-10-01442-t003:** Sensory quality of ‘Florida Radiance’ strawberries from conventional and reduced fungicide disease control treatments held under a steady temperature regime at 1.5 °C and 85% RH or exposed to simulated supply chain conditions ^1^.

Treatment	Appearance ^2^	Texture ^2^	Flavor ^2^	Overall Liking
Harvest 1				
Conventional: control	6.35 ± 0.17 c ^3^	6.12 ± 0.18 b	5.81 ± 0.17 b	5.91 ± 0.17 b
Conventional: supply chain	6.70 ± 0.16 b	6.17 ± 0.17 b	5.79 ± 0.16 b	5.96 ± 0.16 b
Reduced: control	7.22 ± 0.14 a	6.97 ± 0.14 a	6.25 ± 0.17 a	6.54 ± 0.15 a
Reduced: supply chain	6.96 ± 0.15 ab	6.40 ± 0.17 b	5.95 ± 0.18 ab	6.08 ± 0.18 b
Harvest 2				
Conventional: control	6.57 ± 0.16 ab	6.63 ± 0.16 a	6.09 ± 0.17 a	6.24 ± 0.17 a
Conventional: supply chain	6.25 ± 0.17 b	6.11 ± 0.16 b	5.65 ± 0.18 b	5.83 ± 0.17 b
Reduced: control	6.71 ± 0.17 a	6.60 ± 0.16 a	5.75 ± 0.19 ab	6.00 ± 0.18 ab
Reduced: supply chain	6.49 ± 0.16 ab	6.34 ± 0.16 ab	5.73 ± 0.17 ab	6.00 ± 0.16 ab

^1^ Sensory quality was evaluated at the consumer level after washing the fruit (see Table 1); ^2^ 1 = dislike extremely, 5 = neither like nor dislike, 9 = like extremely; ^3^ Letters after averages denote significant differences (*p* < 0.05) between treatments based on Tukey’s HSD test; averages followed by the same letter are not significantly different.

**Table 4 foods-10-01442-t004:** Residual fungicide of ‘Florida Radiance’ strawberries from conventional and reduced fungicide disease control treatments held under a steady temperature regime at 1.5 °C and 85% RH or exposed to simulated supply chain conditions ^1^.

Treatment	Fludioxonil (ppb)	Cyprodinil (ppb)	Pyraclostrobin (ppb)	Captan (ppb)
Harvest 1				
Conventional				
Control: not washed	18 ± 1 c	20 ± 1 c	ND ^2^	4634 ± 36 b
Control: washed	10 ± 1 e	9 ± 1 f	ND	3752 ± 757 c
Supply chain: not washed	28 ± 3 a	35 ± 1 a	20 ± 1	4878 ± 428 b
Supply chain: washed	21 ± 0 b	29 ± 1 b	ND	3131 ± 117 c
Reduced				
Control: not washed	16 ± 1 cd	19 ± 0 c	ND	1634 ± 134 d
Control: washed	9 ± 0 e	12 ± 1 e	ND	857 ± 61 e
Supply chain: not washed	8 ± 0 e	8 ± 1 f	ND	6316 ± 213 a
Supply chain: washed	14 ± 1 d	16 ± 0 d	ND	522 ± 22 e
Harvest 2				
Conventional				
Control: not washed	10 ± 1 b	8 ± 1 bc	ND.	5589 ± 289 a
Control: washed	5 ± 0 e	7 ± 0 c	ND.	3363 ± 30 bc
Supply chain: not washed	6 ± 0 d	7 ± 0 c	ND.	6535 ± 148 a
Supply chain: washed	3 ± 0 f	6 ± 0 d	ND.	3572 ± 264 bc
Reduced				
Control: not washed	9 ± 1 c	7 ± 0 c	ND.	5683 ± 1205 a
Control: washed	5 ± 0 e	6 ± 0 d	ND.	2416 ± 78 cd
Supply chain: not washed	24 ± 1 a	26 ± 0 a	ND.	3849 ± 416 d
Supply chain: washed	5 ± 0 e	8 ± 0 b	ND.	1561 ± 102 b

^1^ Analysis of residual fungicides was conducted at the end of the supply chain simulation: at the consumer level before (not washed) and after washing (washed) the fruit. Data correspond to the means of ± standard error of three biological replicates of 12 strawberries each. Different small letters in the same column show a significant difference (*p* ≤ 0.05) between treatments within each harvest; ^2^ N.D.—Not detected.

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
