# Peer review of "Sensory and Physicochemical Quality, Residual Fungicide Levels and Microbial Load in ‘Florida Radiance’ Strawberries from Different Disease Control Treatments Exposed to Simulated Supply Chain Conditions"

_foods, 2021, doi:10.3390/foods10071442_

Round 1

Reviewer 1 Report

The title of this study is: Sensory and Physicochemical Quality, Residual Fungicide Levels and Microbial Load in Strawberries from Different Disease-Control Treatments Exposed to Simulated Supply Chain Conditions. In this research work, the Authors determine the impact of supply chain conditions on the sensory quality, physicochemical attributes, fungicide residues, and microbial load on strawberries grown conventionally or under the reduced-fungicide application.

I commented on the manuscript and the comments are presented below:

The Introduction to the study is clearly. The purpose of the research was clearly stated.

The Methods section provides the reader with enough information to repeat the experiments conducted. Only the basic statistical analysis was used to describe the differences. Have the Authors attempted to use other more comprehensive statistical analyzes, e.g. principal components analysis of PCA? With such a large number of parameters tested, which may affect the characteristics examined, the Principal Component Analysis (PCA) should be used to results analyzed. More advanced statistical analysis should be performed. The use of advanced statistical methods to fully describe the relationship between the parameters studied and the aspects of the research work carried out in the presented manuscript. You can determine the strength of the influence of a particular parameter on the variance of the system. At the same time, correlation relationships between the determined parameters can be determined.

For the most part the Results and Discussion sections is well structured.

The Conclusions section is sparse and does not contain information obtained after conducting experiments, only known information.

I have a question regarding the collection of research material. The authors conclude that "Disease-control systems that use a lower numbers of fungicide sprays during the strawberry production season are a promising alternative to conventional agricultural practices. Strawberries grown under reduced-fungicide applications have better or similar overall quality than their counterparts grown under conventional methods, but they also have much lower residual fungicides. Also, regardless of the disease-control treatment used, the quality of strawberries is better maintained when the fruit is handled at a constant optimum temperature throughout the supply chain, from the field to the consumer." There are two different statements here because they either have better quality or something similar. Also in the sentences "Our study also shows the importance of washing strawberries before consumption to reduce fungicide and microbial loads on the fruit. Overall, using reduced-fungicide systems combined with proper handling during the supply chain helps reduce strawberry waste while providing the consumer fruit with better quality." This information is already common knowledge. In addition, there is a lack of information about the yield losses in the field between strawberries grown conventionally and those grown with spraying with reduced fungicide content. Only good quality strawberries were taken for the research, as I noticed there is no information about yield losses between crops. The fact that strawberries grown with a lower dose of fungicides are better for the health of the consumer is known without performing the presented tests. Conclusions should be redrafted taking into account the results of the studies performed and described in this manuscript, rather than focusing on the descriptive qualitative information of the fruit.

The literature used is appropriate.

Author Response

Reviewer #1

The title of this study is: Sensory and Physicochemical Quality, Residual Fungicide Levels and Microbial Load in Strawberries from Different Disease-Control Treatments Exposed to Simulated Supply Chain Conditions. In this research work, the Authors determine the impact of supply chain conditions on the sensory quality, physicochemical attributes, fungicide residues, and microbial load on strawberries grown conventionally or under the reduced-fungicide application.

I commented on the manuscript and the comments are presented below:

The Introduction to the study is clearly. The purpose of the research was clearly stated.

The Methods section provides the reader with enough information to repeat the experiments conducted. Only the basic statistical analysis was used to describe the differences. Have the Authors attempted to use other more comprehensive statistical analyzes, e.g. principal components analysis of PCA? With such a large number of parameters tested, which may affect the characteristics examined, the Principal Component Analysis (PCA) should be used to results analyzed. More advanced statistical analysis should be performed. The use of advanced statistical methods to fully describe the relationship between the parameters studied and the aspects of the research work carried out in the presented manuscript. You can determine the strength of the influence of a particular parameter on the variance of the system. At the same time, correlation relationships between the determined parameters can be determined.

RE: Thank you for suggesting an alternative approach to performing the statistical analysis. Although we understand that PCA could have been performed, we believe that the methodology used is appropriated as is provides a reliable way to determine the significance between the results from each experiment.   

For the most part the Results and Discussion sections is well structured.

The Conclusions section is sparse and does not contain information obtained after conducting experiments, only known information.

RE: Rewritten

I have a question regarding the collection of research material. The authors conclude that "Disease-control systems that use a lower numbers of fungicide sprays during the strawberry production season are a promising alternative to conventional agricultural practices. Strawberries grown under reduced-fungicide applications have better or similar overall quality than their counterparts grown under conventional methods, but they also have much lower residual fungicides. Also, regardless of the disease-control treatment used, the quality of strawberries is better maintained when the fruit is handled at a constant optimum temperature throughout the supply chain, from the field to the consumer."

There are two different statements here because they either have better quality or something similar.

RE: Correct, there are two statements: the first regarding the overall quality of reduced-fungicide fruit compared to conventional; the second regarding the effect of handling temperature on the overall quality of strawberries.

Also in the sentences "Our study also shows the importance of washing strawberries before consumption to reduce fungicide and microbial loads on the fruit. Overall using reduced-fungicide systems combined with proper handling during the supply chain helps reduce strawberry waste while providing the consumer fruit with better quality." This information is already common knowledge.

RE: We agree that this information might be common knowledge as practically everyone knows that washing with water cleans. However, this knowledge seems to be mostly based on general assumptions as we did not find any publications showing the effect of washing with tap water on the levels of residual fungicides and microbial load on strawberries. Thus, we believe our study provides scientific-based knowledge that validates common knowledge.

In addition, there is a lack of information about the yield losses in the field between strawberries grown conventionally and those grown with spraying with reduced fungicide content. Only good quality strawberries were taken for the research, as I noticed there is no information about yield losses between crops. The fact that strawberries grown with a lower dose of fungicides are better for the health of the consumer is known without performing the presented tests.

RE: In this study, we did not perform tests on yield loss between the two types of disease-control treatment. However, members of our team have published a study where they show that the model used to determine when fungicide should be applied can be used as a disease-forecasting system and greatly reduce the number of applications without loss of disease control or yield (Please see reference number 26. MacKenzie, S.J.; Peres, N.A. Use of leaf wetness and temperature to time fungicide applications to control Botrytis fruit rot of strawberry in Florida. Plant Dis. 2012, 96, 529-536, doi:10.1094/PDIS-03-11-0181.)

Conclusions should be redrafted taking into account the results of the studies performed and described in this manuscript, rather than focusing on the descriptive qualitative information of the fruit.

RE: Same comment as above. Conclusions section was rewritten.

The literature used is appropriate.

Reviewer 2 Report

Overall the authors present a careful study to investigate the effects of a reduced fungicide regime on a particular variety of strawberries finding interesting results. In addition to being of interest to a scientific audience working in this field, it could also have impact on agricultural practices.

That being said, there are some significant experimental design issues which limit the confidence of findings.

Firstly, a single cultivar of strawberry was used. Cultivar variety of many crop species are a determinant factor for a number of quality and organoleptic properties. Secondly, the supply chain design (and the variety used ) are definitely Florida centric, although strawberries are cultivated commercially in a number of regions. I think the results reported here are valid, but any conclusions and possibly the title and abstract should be modified to indicate the specific nature of this experimental design.

The second major issue is based on treatment gradient. It is quite difficult to have high confidence in the observed results when a simple conventional vs reduced system is examined. I feel that this work could have had a much higher meaning if different levels (concentrations) of treatment were used. 

Although the work seems to be well conducted, these two major issues makes the results and impact fairly narrow.

Major comments

Please state if the individuals who performed the analysis of appearance, texture, flavour and overall liking were aware of the treatment groups of the samples? For example, did the testers know that they were tasting a reduced: control strawberry or was it a blind analysis?

Minor comments

Introduction

Line 49: Can this sentence be made clearer? Are the authors trying to state that above an absolute temperature of 1C vitamin C and total phenolics can be reduced, or, fluctuations (delta +/- 1C) leads to this reduction?

Methods –

Line 221 – HPLC with mass spectrometry should be HPLC coupled to a mass spectrometer.

Table 3 – Why is there a single superscript 3 after conventional: supply chain / appearance value of 6.7?

3.4 – What is the basis for the observation that acidity decreased overtime? In this work or other work, has this been linked to microbial activity?

Table 4 – please convert values to ppb

3.6 -  Many microbes invert sucrose into its components fructose and glucose. What is interesting is that although a reduction of sucrose is observed, there is not an increase in glucose and fructose, meaning that they are likely being consumed as an energy source? Could you provide a plausible explanation of what is happening to the sugars?

Author Response

Reviewer #2

Overall the authors present a careful study to investigate the effects of a reduced fungicide regime on a particular variety of strawberries finding interesting results. In addition to being of interest to a scientific audience working in this field, it could also have impact on agricultural practices.

That being said, there are some significant experimental design issues which limit the confidence of findings.

Firstly, a single cultivar of strawberry was used. Cultivar variety of many crop species are a determinant factor for a number of quality and organoleptic properties.

RE: We agree that organoleptic characteristics can vary depending on the cultivar. However, the main objective of this study was to show the effect of disease-control treatments and supply chain conditions on the overall quality of strawberries rather than the differences between the organoleptic qualities of different cultivars. Besides, the cultivar used in this study is one of the major commercial standards in Florida and it is also grown in Europe. Furthermore, the complexity of the experimental design and the amount of work committed to this study did not allow for analysis of multiple cultivars. However, in a previous study we used different strawberry cultivars grown under different disease control treatments and showed that although there were variations between cultivars, the trend was similar (Please see reference #15 Abountiolas, M.; Kelly, K.; Yagiz, Y.; Li, Z.; Mahnken, G.; Borejsza-Wysocki, W.; Marshall, M.; Sims, C.A.; Peres, N.; do Nascimento Nunes, M.C Sensory Quality, Physicochemical Attributes, Polyphenol Profiles, and Residual Fungicides in Strawberries from Different Disease-Control Treatments. J. Agric. Food Chem. 2018, 66, 6986-6996, doi:10.1021/acs.jafc.8b02153.)

Secondly, the supply chain design (and the variety used ) are definitely Florida centric, although strawberries are cultivated commercially in a number of regions. I think the results reported here are valid, but any conclusions and possibly the title and abstract should be modified to indicate the specific nature of this experimental design.

RE: We agree that strawberries are cultivated in many regions of the world and that each region might have different cultivars adapted to specific environmental conditions, as well as specific supply chain scenarios. The argument could be applied to any study dealing with strawberries or other crops, but one needs to be realistic and be aware that it is impossible to replicate every single possible world scenario that would consider every single strawberry cultivar and supply chain scenario. Most published studies use plant material available locally and still provide valuable scientific information worldwide. We believe that it is well explained in the Material and Methods section the experimental design and why this cultivar and supply chain scenario was used.

The second major issue is based on treatment gradient. It is quite difficult to have high confidence in the observed results when a simple conventional vs reduced system is examined. I feel that this work could have had a much higher meaning if different levels (concentrations) of treatment were used. 

RE: The disease-control treatments used here are based on current commercial practices. The “treatment gradients” were previously studied (Please see reference #15 Abountiolas, M.; Kelly, K.; Yagiz, Y.; Li, Z.; Mahnken, G.; Borejsza-Wysocki, W.; Marshall, M.; Sims, C.A.; Peres, N.; do Nascimento Nunes, M.C Sensory Quality, Physicochemical Attributes, Polyphenol Profiles, and Residual Fungicides in Strawberries from Different Disease-Control Treatments. J. Agric. Food Chem. 2018, 66, 6986-6996, doi:10.1021/acs.jafc.8b02153.)

Although the work seems to be well conducted, these two major issues makes the results and impact fairly narrow.

RE: We do not agree that the impact of this study is narrow. The results show very clearly that reducing the number of fungicides-sprays applied to strawberries is important and beneficial for many reasons discussed in the manuscript. Besides, the methodology used can be applied elsewhere with modifications to adapt to local environmental conditions and agricultural practices.

Major comments

Please state if the individuals who performed the analysis of appearance, texture, flavour and overall liking were aware of the treatment groups of the samples? For example, did the testers know that they were tasting a reduced: control strawberry or was it a blind analysis?

RE: This was a blind sensory analysis test using a consumer panel. It is explained in the M&M-Sensory Analysis section that each treatment was coded by a three-digit random number, meaning that the panelist did not know the treatments.

Minor comments

Introduction

Line 49: Can this sentence be made clearer? Are the authors trying to state that above an absolute temperature of 1C vitamin C and total phenolics can be reduced, or, fluctuations (delta +/- 1C) leads to this reduction?

RE: Rewritten.

Methods –

Line 221 – HPLC with mass spectrometry should be HPLC coupled to a mass spectrometer.

RE: Corrected.

Table 3 – Why is there a single superscript 3 after conventional: supply chain / appearance value of 6.7?

RE: Thank you for catching that up. The superscript should have been placed after the letter following the first average on the table. In the table footnote the explanation is given.  

3.4 – What is the basis for the observation that acidity decreased overtime? In this work or other work, has this been linked to microbial activity?

RE: In general, reduction in acidity levels is associated with the normal postharvest metabolism of the fruit. As the fruit is detached from the mother plant, fruit reserves are used to maintain its metabolic activity, namely respiration. That leads to consumption of organic acid and sugars until eventually the reserves are depleted and the fruit senesces. Microbial activity, mostly fungal decay, can contribute to an increase in respiration rate and a faster decline in energy sources.

Table 4 – please convert values to ppb

RE: ppm converted to ppb. However, the reason why the data was expressed in ppm was to be presented in the same units reported by USEPA. We also converted the units in the manuscript Discussion – section 4.5. (Please see reference #61 61.             USEPA. United States Environmental Protection Agency. Electronic Code of Federal Regulations 40 CFR Part 180. 2018 [cited 2018 March 18, 2018]; Available from: https://www.ecfr.gov/cgi-bin/text-idx?node=pt40.24.180.26.180_13)

3.6 -  Many microbes invert sucrose into its components fructose and glucose. What is interesting is that although a reduction of sucrose is observed, there is not an increase in glucose and fructose, meaning that they are likely being consumed as an energy source? Could you provide a plausible explanation of what is happening to the sugars?

RE: Like what happened with organic acids, sugars are also being used as a source of energy during respiratory metabolism. As the fruit is detached from the mother plant, fruit reserves are used to maintain its metabolic activity, namely respiration. That leads to consumption of organic acid and sugars until eventually the reserves are depleted and the fruit senesces.

We added the following sentence in the Discussion -4.4 section: “Decrease in sugar contents results from the normal metabolic activity after the fruit being detached from the mother plant. The increase in the overall rate of metabolic reactions (e.g., glycolysis) with increased time-temperature exposure contributes to the breakdown of sugars, depleting fruit reserves.”

Reviewer 3 Report

The authors studied the physicochemical properties, sensory quality, residual fungicide levels, and microbiological quality of strawberries after conventional or reduced fungicide treatment. They reported that strawberries grown under reduced-fungicide conditions had better or similar overall quality than those grown under conventional conditions, with much lower residual fungicides. Overall, this is a well-organized and well-written research article.

The data should be interpreted more statistically in the Abstract and Results. If there were significant differences, P<0.05 should be added. If not, P>0.05 should be added.

It is important to note what kinds of fungicide were studied in this study in the Abstract.

SD values should be provided to the data in Table 3.

Standard error bars should be added to the data in Figures 1-6.

It would be easier to read if you could use days for X-axis instead of hours.

Author Response

Reviewer #3

The authors studied the physicochemical properties, sensory quality, residual fungicide levels, and microbiological quality of strawberries after conventional or reduced fungicide treatment. They reported that strawberries grown under reduced-fungicide conditions had better or similar overall quality than those grown under conventional conditions, with much lower residual fungicides. Overall, this is a well-organized and well-written research article.

The data should be interpreted more statistically in the Abstract and Results. If there were significant differences, P<0.05 should be added. If not, P>0.05 should be added.

RE: Statistic (p values) were added in the results when applicable. However, to avoid increasing the length of the manuscript we opted by not adding p values on the abstract assuming that if there were differences between treatments for specific attributes, those are statistically significant with p values <0.05.

It is important to note what kinds of fungicide were studied in this study in the Abstract.

RE: The types of fungicides were added to the abstract. However, the length of the abstract was increased to more than 200 words.

SD values should be provided to the data in Table 3.

RE: SD included in Table 3.

Standard error bars should be added to the data in Figures 1-6.

RE: SE bars added to Figures 1-6.

It would be easier to read if you could use days for X-axis instead of hours.

RE: Hours were converted to days. Table 2 was updated to reflect the changes as well as the graphs. Hours were changed to days throughout the manuscript.

Round 2

Reviewer 1 Report

The authors referred to the comments from the previous review for the manuscript titled:  Sensory and Physicochemical Quality, Residual Fungicide Levels and Microbial Load in Strawberries from Different Disease-Control Treatments Exposed to Simulated Supply Chain Conditions. I accept explanations. In the future, I suggest using more precise describing relationships between the parameters studied. 

Author Response

Dear Reviewer,

I appreciate very much your input and I will definitely look into new approaches to statistical analysis. Thank you!

Reviewer 2 Report

The authors addressed my minor concerns regarding spelling, sentence clarity and some table/numerical formatting issues.

However they did not address any of the major comments I had in a meaningful way.  I understand that adding additional cultivars, different fungicide concentrations are not possible without significant work, and likely will yield similar results. However, this should be made clear in the paper, prominently. 

In short, this work was on a single strawberry cultivar yet throughout the document, this is not clear and it is implied that this is the case for all strawberries. There is no evidence in this paper that this is the case.

At the very least, the abstract should be altered to indicate that the study was completed on a single strawberry variety. In the conclusions, it should be stated that additional experiments with different varieties would be add value to the overall findings.

Author Response

Dear Reviewer,

We have added more details to indicate that only one strawberry cultivar was used.

Reviewer 3 Report

N.A.

Author Response

Dear Reviewer,

Thank you for accepting the manuscript after our revision.